# Hyperammonemic Encephalopathy in a Patient with Pancreatic Neuroendocrine Tumor and Portosystemic Shunt

**DOI:** 10.3390/diagnostics13030572

**Published:** 2023-02-03

**Authors:** Marcel Zorgdrager, Frans J. C. Cuperus, Robbert J. de Haas

**Affiliations:** 1Department of Radiology, University Medical Centre Groningen, University of Groningen, 9713 GZ Groningen, The Netherlands; 2Department of Gastro-Enterology and Hepatology, University Medical Centre Groningen, University of Groningen, 9713 GZ Groningen, The Netherlands

**Keywords:** pancreatic neuroendocrine tumor (PNET), portosystemic shunt, hyperammonemia

## Abstract

Hyperammonemia can lead to encephalopathy and may be accompanied by a diagnostic dilemma. Imaging as well as biochemical analyses are the cornerstone for identifying possible underlying causes such as severe liver disease or urea cycle defect. We report a case of a patient that presented with neurological deficits based on hyperammonemia in the presence of a large pancreatic neuroendocrine tumor (PNET) and portosystemic shunts in the liver. Prior cases are rather scarce, and the exact mechanism is not fully understood. The case illustrates the added value of a multimodality imaging approach in patients presenting with hyperammonemia-induced encephalopathy.

## 1. Introduction

Hyperammonemia can lead to encephalopathy and symptoms may vary between mild encephalopathy and coma or even death, depending on the serum concentration. It is most commonly seen in severe liver disease and, less often, in urea cycle or mitochondrial disorders. Urea cycle disorders should be suspected in cases of hyperammonemia with normal blood glucose level but without metabolic acidosis or impaired liver function tests [1]. Biochemical tests and magnetic resonance imaging (MRI) of the brain as well as an abdominal computed tomography (CT) scan should be performed to differentiate between possible causes [2]. Hyperammonemia in the context of a neuroendocrine tumor in the pancreas without extensive liver metastases or concomitant severe liver disease is rare. However, it has been reported previously, and the most likely theory is the presence of portosystemic shunts in which the liver is bypassed [3,4,5,6,7,8,9,10]. Extensive liver metastases of neuroendocrine tumors may induce encephalopathy. 

We report a case of a patient presenting with a hyperammonemia-induced encephalopathy most likely due to a pancreatic neuroendocrine tumor and portosystemic shunts in the liver. 

## 2. Case Report

A 36-year-old man was admitted to our hospital with neurological deficits after resection of a diffuse astrocytoma (WHO grade II). He had a history of Usher syndrome type II with microcephaly and psychomotor retardation. The patient presented with drowsiness, apraxia, urinary incontinence, and repeated insults two days after craniotomy. An MRI showed minimal intracranial hematoma, which was surgically evacuated. However, neurological symptoms persisted. Signs of encephalopathy were seen on electroencephalogram, but no epileptic activity was detected. Additional metabolic analysis revealed a raised ammonia of 185 μmol/L. Potential causes of a urea cycle disorder and cirrhosis were ruled out. Total body CT and MRI of the pancreas showed a enlarged hyperenhancing pancreas with multiple calcifications and a restricted diffusion. Furthermore, portosystemic shunts (Park type IV) in liver segments III and VIII were detected. A Ga-68 DOTATOC positron emission tomography (PET) showed a diffuse increased uptake of Ga-68 in the enlarged pancreas (Figure 1). 

Fine-needle aspiration biopsy was performed twice and ultimately confirmed a pancreatic neuroendocrine tumor (PNET) which was positive for synaptophysin, chromogranin A, CK8/18, and CD56. No reliable Ki-67 expression or mitotic count could be evaluated due to the small number of tumor cells in the aspirate. Additional biochemical analysis showed a raised serum chromogranin A (CgA) of 591 μg/L (ref. 20–100 μg/L) as well as an increased serum serotonin of 7.9 nmol/10^9^ trombocytes. This further supported the diagnosis of a neuroendocrine tumor. Both plasma 5-hydroxyindoleacetic acid (5HIAA) and 24-h urine 5HIAA were within normal range. A follow-up CT after 5 months showed stable disease. 

The hyperammonemia-induced encephalopathy was successfully treated with lactulose as the maintenance medication. Treatment options such as coiling the portosystemic shunts, a total pancreatectomy, and somatostatin analogues were considered. However, due to the patient’s condition, medical history, the fact of it being most likely a low-grade tumor, the possible low benefit of somatostatins, and the wishes of the family, a palliative comfort and care approach was chosen.

## 3. Discussion

Ammonia is a nitrogenous residual product of protein catabolism and is metabolized to glutamine largely by the liver and in a lesser degree in astrocytes and skeletal muscle. Raised concentration can occur due to overproduction, impaired liver function, and portosystemic shunting. Hyperammonemia is most commonly caused by chronic liver disease or extensive metastatic liver disease with portosystemic shunts. Inherited urea cycle defects, such as ornithine transcarbamylase deficiency (OTCD), are another important cause and manifest themselves most often in childhood, but late onset urea cycle defects in adults do occur [1,11]. An overview of acquired causes is shown in Table 1.

Neuroendocrine neoplasia (NEN) are tumors with neuroendocrine differentiation in which biomarkers are expressed and that have an impact on the secretion of normal neuroendocrine cells (epithelial types) or neurons (paraganglioma) [13]. It concerns functioning tumors in up to 30% of cases causing hypersecretion of certain hormones that clinically can result in a carcinoid syndrome or specific clinical syndromes such as Cushing and Zollinger–Ellison [14]. There is an increased risk of development of PNEN in patients with multiple endocrine neoplasia syndrome type 1 (MEN1) and von Hippel-Lindau syndrome (VHL). A well-differentiated NEN is classified as a neuroendocrine tumor (NET) and can be further graded into grades 1–3, based on the mitotic count and/or proliferation index (Ki-67). In pancreatobiliary- and gastrointestinal-located tumors, grade one are low-grade tumors with a Ki67-index <3% and/or <2 mitoses/mm^2^. Grade two are intermediate NETs with a Ki67-index of 3–20% and/or 2–20 mitoses/mm^2^. Grade three are high-grade NETs with a Ki67-index >20% and/or >20 mitoses/mm^2^. A poorly differentiated tumor is considered to be a neuroendocrine carcinoma (NEC) and can be divided in large or small cell types. NECs are by definition high grade tumors (grade three). The Ki67 index and mitotic counts are the cornerstone in grading NENs. Additional biomarkers on immunohistochemical analyses consist of synaptophysin, chromogranin A and insulinoma-associated protein 1 (INSM1). Keratins can be used to differentiate between epithelial and non-epithelial NEN. NENs are known to metastasize primarily to the liver and lymph nodes.

Diffuse enlargement of the pancreas due to the diffuse or multifocal localization of PNET is rare, but has been reported previously and may resemble chronic pancreatitis, especially in the presence of coarse calcifications (Table 2). However, signs of parenchymal atrophy and marked ductal dilatation, often with intraductal calcifications, are more frequently present in chronic pancreatitis.

In imaging, low and intermediate grade PNETs are more frequently well-defined small lesions (<3 cm), with homogeneous hyperenhancement in CT and MRI in both the arterial and portovenous phases. In MRI, the lesions show a typically low signal intensity in T1-weighted imaging and a high signal intensity in T2-weighted imaging. Restricted diffusion is not always seen in low-grade types. Higher-graded or poorly differentiated NENs are often larger (>3 cm), show low T1 and an intermediate to low T2 signal intensity, with a more heterogeneous enhancement pattern. They often do show restricted diffusion as well as calcifications and signs of necrosis or cystic changes [14]. High-grade tumors can show extramural venous invasion, which is less often seen in conventional pancreatic adenocarcinoma. On the other hand, intraductal spread is more often observed in pancreatic adenocarcinoma. Other mimickers of PNET are hypervascular metastasis, splenule, small microcystadenoma, intraductal papillary mucinous neoplasm, and gastrointestinal stromal tumors.

PNEN show somatostatin receptor expression in up to 80% [14]. A PET/CT with a ^68^Ga-DOTATATE tracer has a sensitivity of 81–100% and a specificity of 90–100% in the detection of NETs. Poorly differentiated or tumors with higher grade (G3) show a somatostatin receptor expression less often. PET/CT with ^18^F-FDG tracer is known to have a higher sensitivity in these cases as compared to DOTATATE PET/CT [14]. In our case diffuse intense uptake of ^68^Ga-DOTATATE was observed, and therefore, no further ^18^F-FDG-PET was performed.

Pathological examination in our case showed a well differentiated PNET, and this is in line with previous cases of diffuse localized PNET (Table 2). Most likely these are non-functional and slow-growing tumors and therefore remain asymptomatic for a long period of time. However, surgery is known to provoke carcinoid crisis in NEN in up to 20% due to the release of vasoactive hormones [15]. It is unclear whether brain surgery in our patient could have been a triggering factor of the PNET to manifest clinically.

There are a few cases described previously of NENs associated with hyperammonemia (Table 3). In only two cases of PNET with hyperammonemia, no hepatic tumor localization was described, and this is in accordance with our patient [3,8]. Our case supports the theory of either intratumoral or portosystemic shunting resulting in a bypass of the liver. Due to the size of the PNET and the presence of portovenous hepatic shunts, portal hypertension was likely to be present in our case. Most previous cases described the presence of portal hypertension and in one case, intratumoral arterioportal shunting was actually visualized in liver metastases during transarterial chemoembolization (TACE) [6]. In this case, the encephalopathy resolved after treatment. Similar results were reported in another case in which a hepatic NET was treated with TACE [9]. Hepatic encephalopathy is usually a contraindication for TACE treatment because the embolization of tumor-free liver parenchyma could potentially aggravate encephalopathy. However, hyperammonemic patients with extensive tumor burden of NEN with otherwise no signs of liver failure could potentially be eligible for TACE treatment. In patients with diffuse localized PNET, total pancreatectomy can be considered a curative option. Surgical resection is also preferred in smaller functional or large non-functional PNETs. Peptide receptor radionuclide therapy (PRRT) with ^177^Lu-DOTATATE and somatostatin analogs shows promising results as a palliative treatment for patients with extensive disease [21].

Chemotherapy or radiation therapy are the best options in extensive disease in case of low somatostatin receptor expression. Hyperammonemia can be treated by administering lactulose or rifaximin, and in severe cases hemodialysis can be considered. 

In conclusion, this case illustrates an association between hyperammonemia-induced encephalopathy and PNET without extensive liver metastases or concomitant severe liver disease, in which portosystemic shunting seems to be an important contributing factor. A multimodality approach is mandatory in patients presenting with hyperammonemic encephalopathy in order to assess the presence of extensive liver disease and/or portosystemic shunts. 

## Figures and Tables

**Figure 1 diagnostics-13-00572-f001:**
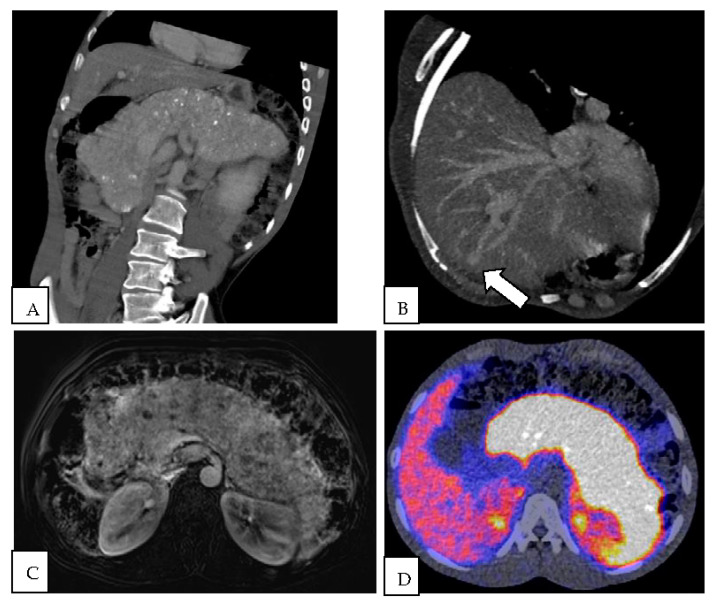
(**A**) Angulated coronal CT image in the portovenous phase showing a diffuse enlarged pancreas with peripheral calcifications. (**B**) CT Maximum Intensity Projection (MIP) reconstruction of a shunt (arrow) between the left portal vein and middle hepatic vein. (**C**) T1-weighted fat-suppressed MRI subtraction image in the arterial phase in transverse direction. Diffuse enhancement of the extremely enlarged pancreas is visible with multiple low intensity foci corresponding to calcifications. (**D**) Fused image of the Gallium-68 DOTATATE scan in transverse direction, showing diffuse increased uptake of the enlarged pancreas.

**Table 1 diagnostics-13-00572-t001:** Causes of acquired hyperammonemia [12].

Cause	Mechanism or Examples
Extensive liver disease due to cirrhosis or metastases	Impaired urea cycle due to damaged hepatocytes and portosystemic shunting
Kidney failure	Excess of ammonia cannot be excreted as urea
Small intestinal bacterial overgrowth	Increased production of ammonia
Urease producing infection/sepsis	Increased production of ammonia, e.g., Proteus, H.Pylori, Nocardia, Klebsiella, Cryptococcus, Mycoplasma
Drugs	Valproic acid, carbamazepine, salicylate intoxication, f-fluorouracil, L-asparaginase
Reye syndrome	Impaired urea cycle, mitochondrial dysfunction

**Table 2 diagnostics-13-00572-t002:** Previous studies reporting diffuse enlarged pancreas based on NEN.

Reference	Differentiation	Grade NEN
Naringrekar et al., 2017 [16]	Unknown	Grade 2 (KI-67: 5%)
Singh et al., 2008 [8]	Well	Unknown
Santes et al., 2017 [17]	Well	Grade 1 (Ki-67: <1%)
Bhargava et al., 2018 [18]	Well	Grade 1 (Ki-67: <2%)
Salahshour et al., 2021 [19]	Well	Grade 1 (Ki-67: <1%)
Yazawa et al., 2011 [20]	Well	Grade 3 (Ki-67: 30–40%)

**Table 3 diagnostics-13-00572-t003:** Previous studies reporting hyperammonemia and portosystemic shunt in the presence of pancreatic and/or hepatic NEN.

Reference	Pancreatic Involvement NET	Hepatic Involvement NET	Hepatic Vascular Pathology
Monardo et al., 2020 [3]	Yes	No	Possible portosystemic shunt.
Broadbridge et al., 2010 [4]	Yes	Yes	Portosystemic shunt due to portal vein thrombosis.
Clinicopathologic conference. 1991 [5]	Yes	Yes	Portosystemic shunt due to tumoral portal vein thrombosis.
Erinjeri et al., 2010 [6]	Yes	Yes	Portosystemic shunt due to arterioportal shunt.
Turken et al., 2009 [7]	No	Yes	No.
Singh et al., 2008 [8]	Yes	No	Atypical hemangioma. Signs of portal hypertension (splenomegaly, engorgement of portal vein and esophageal varices) but no cirrhosis.
Pande et al., 2016 [9]	No	Yes	No.
Vandamme et al., 2012 [10]	Yes	Yes	Mechanical compression of tumor on portal vein.

## Data Availability

Not applicable.

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
