# Peer review of "Hyperammonemic Encephalopathy in a Patient with Pancreatic Neuroendocrine Tumor and Portosystemic Shunt"

_diagnostics, 2023, doi:10.3390/diagnostics13030572_

Round 1

Reviewer 1 Report

After delivering the necessary general information, the authors present a rare case of hyperammonemia-induced encephalopathy due to a pancreatic neuroendocrine tumor and portosystemic shunts in the liver.

The information covers the topic, which is very rare.

The presentation is clear, comprehensive and well documented.

The references are appropriate, up-to-date and contain 20 titles, corresponding to rarity of the case.

I found no self-citations.

The figures (4) are appropriate and mandatory for sustaining the case presention .

The 3 tables offer concentrated information on the topic.

The discussions and conclusions are coherent and connected to the content.

In my opinion the paper fits the journal and the language is correct and understandable.

I recommend the paper to be accepted.

Author Response

We would like to thank reviewer 1 for the kind remarks and recommendation for publication.

Reviewer 2 Report

Dear Authors,

This is an interesting case report.  I enjoyed reading the paper which is well structured and adequate to the topic.

The area of neuroendocrine neoplasia still represents a challenging from a multi-disciplinary point of view.

I only have a few minor observations or suggestions:

1.       Introduction – I suggest you comment that hyperammonemia is rare in neuroendocrine tumors without actual severe hepatic spreading (metastasis) or concomitant liver involvement of other causes (like concomitant viral or alcohol – induced cirrhosis, etc). Otherwise, typical liver involvement in metastatic neuroendocrine neoplasia may complicate in various ways, including encephalopathy.

2.       Case report – Line 42 – Please use “MRI” once the abbreviation has been introduced (at Introduction - line 28)

3.       Case report – Line 78 – The evaluation of “low grade tumor” was based on mitotic count since Ki67 was not available?

4.       Case report – Do you have the assays for blood and/or urinary neuroendocrine markers?

5.       Please provide some references for Table 1

6.       Please use “neuroendocrine” instead of “neuro-endocrine”

7.       Discussion – Line 94 – Typical clinical expression in neuroendocrine neoplasia is carcinoid syndrome

8.       Discussion – Line 96 – Once you introduced the abbreviation, you should use it e.g. NEN”

9.       Discussion – Line 104 – A most important tool at immunohistochemistry report remains the assessment of Ki67 proliferation marker since prior and current grading systems take into consideration the value of Ki67

10.   Conclusion – You should mention a very important aspect “In conclusion, this case illustrates an association between hyperammonemia-induced encephalopathy and pancreatic neuroendocrine tumor in which portosystemic shunting seems to be an important contributing factor” + in the absence of liver metastasis /spreading of the underlying condition.

Thank you

Author Response

Response to Reviewer 2 Comments

  1. Introduction – I suggest you comment that hyperammonemia is rare in neuroendocrine tumors without actual severe hepatic spreading (metastasis) or concomitant liver involvement of other causes (like concomitant viral or alcohol – induced cirrhosis, etc). Otherwise, typical liver involvement in metastatic neuroendocrine neoplasia may complicate in various ways, including encephalopathy.

Response 1: Please provide your response for Point 1. (in red)

We would like to thank reviewer 2 for the comments. For reasons of clarity, we have made some changes to the manuscript concerning the suggested comments. Indeed, it is a very rare case which is not often seen in patients with neuroendocrine tumors without extensive liver metastases. This has been added to the Introduction on page 1.

  1. Case report – Line 42 – Please use “MRI” once the abbreviation has been introduced (at Introduction - line 28)

Response 2: Please provide your response for Point 2. (in red)

This is a good point of the reviewer. We have made changes to the manuscript in which ‘MRI’ is now being used after the introduction.

  1. Case report – Line 78 – The evaluation of “low grade tumor” was based on mitotic count since Ki67 was not available?

Response 3: Please provide your response for Point 3. (in red)

The patient underwent endoscopic evaluation twice. The first time no definite diagnosis could be made. The second time too few tumor cells were aspirated to accurately assess Ki67-index or mitotic count. However, it was positive for synaptophysin, chromogranin A, CK8/18 and CD56. The features of the tumor on initial imaging as well as on follow up scan, were consistent with a well differentiated tumor that is most likely low grade. Because of a palliative setting with comfort and care, no further pathological material were obtained. We made a nuance in the manuscript in which we state that it was most likely a low grade tumor (page 3).   

  1. Case report – Do you have the assays for blood and/or urinary neuroendocrine markers?

Response 4: Please provide your response for Point 4. (in red)

Additional assays of blood and urine were indeed performed and consisted of chromogranin A (CgA), serum serotonin, plasma 5-hydroxyindoleacetic acid (5HIAA) and 24-hour urine 5HIAA. These were added to the manuscript (first paragraph on page 3).

  1. Please provide some references for Table 1

Response 5: Please provide your response for Point 5. (in red)

We have provided the following reference to the table:

  1. Ali and S. Nagalli. Hyperammonemia. [Updated 2022 Aug 8]. In: StatPearls [Internet]. Treasure Island (FL): StatPearls Publishing; 2022 Jan-. Available from: https://www.ncbi.nlm.nih.gov/books/NBK557504/
  2. Please use “neuroendocrine” instead of “neuro-endocrine”

Response 6: Please provide your response for Point 6. (in red)

This has to do with our automatic spell checker which has been somewhat inconsistent. We have changed this to ‘neuroendocrine’ throughout the manuscript.

  1. Discussion – Line 94 – Typical clinical expression in neuroendocrine neoplasia is carcinoid syndrome

Response 7: Please provide your response for Point 7. (in red)

The reviewer has a good point. We have added the mentioned clinical expression to the discussion section (page 3).

  1. Discussion – Line 96 – Once you introduced the abbreviation, you should use it e.g. NEN”

Response 8: Please provide your response for Point 8. (in red)

We made some changes to the manuscript in which the mentioned abbreviations are used after introduction of the abbreviation.

  1. Discussion – Line 104 – A most important tool at immunohistochemistry report remains the assessment of Ki67 proliferation marker since prior and current grading systems take into consideration the value of Ki67

Response 9: Please provide your response for Point 9. (in red)

We agree with the reviewer on this point, and therefore,we have tried to emphasize the value of Ki67 and mitotic counting a bit more in the discussion (page 4).

  1. Conclusion – You should mention a very important aspect “In conclusion, this case illustrates an association between hyperammonemia-induced encephalopathy and pancreatic neuroendocrine tumor in which portosystemic shunting seems to be an important contributing factor” + in the absence of liver metastasis /spreading of the underlying condition.

Response 10: Please provide your response for Point 10. (in red)

Again, we would like to thank the reviewer for the comments and made some significant changes to the manuscript. Concerning this final comment, we altered the conclusion with the proposed nuance of the abcence of liver metastases or liver disease in our patient (page 5).
